# A Functional Network of Novel Barley MicroRNAs and Their Targets in Response to Drought

**DOI:** 10.3390/genes11050488

**Published:** 2020-04-29

**Authors:** Aleksandra Smoczynska, Andrzej M. Pacak, Przemysław Nuc, Aleksandra Swida-Barteczka, Katarzyna Kruszka, Wojciech M. Karlowski, Artur Jarmolowski, Zofia Szweykowska-Kulinska

**Affiliations:** 1Department of Gene Expression, Institute of Molecular Biology and Biotechnology, Faculty of Biology, Adam Mickiewicz University, 61-614 Poznan, Poland; aleksandra.smoczynska@amu.edu.pl (A.S.); apacak@amu.edu.pl (A.M.P.); przemyslaw.nuc@amu.edu.pl (P.N.); swidbar@amu.edu.pl (A.S.-B.); katarzyna.kruszka@amu.edu.pl (K.K.); artur.jarmolowski@amu.edu.pl (A.J.); 2Department of Computational Biology, Institute of Molecular Biology and Biotechnology, Faculty of Biology, Adam Mickiewicz University, 61-712 Poznan, Poland; wojciech.karlowski@amu.edu.pl

**Keywords:** microRNA, barley, novel barley microRNAs, microRNA expression pattern, mRNA targets of novel barley microRNAs, plant response to drought

## Abstract

The regulation of mRNA (messenger RNA) levels by microRNA-mediated activity is especially important in plant responses to environmental stresses. In this work, we report six novel barley microRNAs, including two processed from the same precursor that are severely downregulated under drought conditions. For all analyzed microRNAs, we found target genes that were upregulated under drought conditions and that were known to be involved in a plethora of processes from disease resistance to chromatin–protein complex formation and the regulation of transcription in mitochondria. Targets for novel barley microRNAs were confirmed through degradome data analysis and RT-qPCR using primers flanking microRNA-recognition site. Our results show a broad transcriptional response of barley to water deficiency conditions through microRNA-mediated gene regulation and facilitate further research on drought tolerance in crops.

## 1. Introduction

Throughout the course of evolution, plants have developed various mechanisms to alter the expression of genes governing physiological processes in response to changing environmental conditions. MicroRNAs (miRNAs) represent the main class of gene expression regulators that are important in shaping these processes.

MicroRNAs belong to a class of small non-coding RNAs that are usually 21 nt in length. Plant microRNA genes (*MIRs*) are transcribed by RNA polymerase II, and primary *MIR* transcripts (pri-miRNAs) fold into a hairpin structure containing microRNA and its complementary partner, passenger strand- microRNA*. Hairpin structure is further processed by the RNase III enzyme DICER LIKE I (DCL1), which cleaves longer transcripts (pri-miRNA) to shorter pre-miRNAs [1,2]. Subsequent cleavages lead to the release of a miRNA/miRNA* duplex that is further methylated by Hua Enhancer 1 (HEN1) and exported to the cytoplasm through the activity of the HASTY protein, where, in most cases, microRNA* is degraded [3,4,5]. However, other studies have shown that *hst* mutants exhibit a decreased accumulation of some, but not all, miRNAs, which suggests the existence of HST-independent plant microRNA export systems [4]. MicroRNAs are incorporated into the multi-protein RNA-induced silencing complex (RISC) and guide RISC to a target mRNA based on nucleotide complementarity. The core component of the RISC complex is the ARGONAUTE1 (AGO1) protein, which mediates the cleavage of mRNA. The cleavage site within mRNA is located between the residues paired to nucleotides 10 and 11 nt counting from the 5′ end of the miRNA sequence [6,7]. Recently, it has been suggested that microRNAs are loaded onto the AGO protein in the nucleus because AtAGO1 contains functional and conserved nuclear localization and nuclear export signals (NLS and NES, respectively) that enable it to shuttle between the cytoplasm and nucleus. Furthermore, nuclear AGO1 with a mutated NES signal contains the same 2′-O-methylated microRNA cohort as its nucleo-cytosolic counterpart but interacts relatively more efficiently with the loading chaperone HSP90 [8,9]. Studies have revealed that microRNA-mediated regulation can also occur via translation inhibition, as shown for the first time in the case of Arabidopsis microRNA172 [10,11,12]. However, this mechanism seems to be less prevalent in plants than in animals [13].

The importance of plant microRNA-mediated regulation in response to various environmental stresses has been extensively studied [14,15,16].

For example, microRNA396 is involved in the heat–stress response in plants, and it targets members of the WRKY transcription factor family. Transgenic Arabidopsis lines overexpressing the microRNA396-resistant version of *WRKY6* displayed hypersensitivity to high temperatures [17]. MicroRNA393 showed induced expression in wheat in response to salinity stress [18]. What is more, in wheat microRNA1119 was shown to be essential in drought response as its expression was gradually increased over 48 h of stress, which was coupled with decreased expression of target mRNAs. Transgenic tobacco lines overexpressing microRNA1119 from wheat display higher drought tolerance through improved reactive oxygen species (ROS) homeostasis in the cell [19]. Arabidopsis transgenic plants overexpressing microRNA393 exhibited enhanced salt tolerance [20]. In barley, microRNA156 and microRNA6213 were upregulated, while microRNA168, microRNA444, and microRNA5048 were downregulated in response to salinity stress [21]. Moreover, microRNA171, which targets the MYB family of transcription factors, was found to be upregulated by salinity stress in barley, wheat and Arabidopsis [18,21], which suggests that these mechanisms may be evolutionarily conserved between monocots and dicots. Gao et al. reported the upregulation of microRNA444a in the roots and leaves of wheat in response to nitrogen deficiency. The overexpression of its orthologue, microRNA444a, in tobacco improved plant growth and biomass, the N content, photosynthetic parameters, and antioxidant enzymatic activities under N-deficiency conditions. These results suggested that microRNA444a acts as a regulator of nitrate transporters [22]. MicroRNA399 is involved in the P-starvation response and was found to be upregulated in both barley and wheat under these conditions [23,24]. The target of microRNA399 is mRNA encoding the enzyme phosphate 2 (PHO2), and the downregulation of *PHO2* coupled with the upregulation of these miRNAs was observed [23]. Furthermore, the overexpression of this microRNA in tomato resulted in increased Pi uptake [24]. These observations suggest that microRNA399 is a good candidate for the manipulation of the Pi uptake pathway in wheat and barley.

As the human population grows, the requirements for crop yields increase. There is an emphasis on drought-related research in cereals because drought stress impacts grain yield by reducing the number of tillers, spikes, grains per plant and individual grain weight [25]. Combating the loss of production due to drought stress is especially important for barley because it is grown worldwide and is an economically important cereal, ranking 4th in global production, that is used in the food industry, in beer production and as a fodder in agriculture [26]. Also, barley is a winter cereal cultivated in extreme drought stress conditions for instance in North Africa [26]. Additionally, due to its diploid genome, barley also serves as an important model in genetic studies. Knowledge of the molecular basis of barley cultivar stress tolerance and adaptation is essential to develop plants with improved stress tolerance. To date, many drought-responsive microRNAs have been reported in barley (*Hordeum vulgare*) [27].

MicroRNA156a, microRNA166, microRNA171, and microRNAR408 were found to be upregulated in barley leaves under dehydration-stress conditions [28]. Recently, 31 barley microRNAs were detected in the Golden Promise cultivar treated with drought, 13 of which were significantly downregulated, while one microRNA (hvu-microRNA5049b) was significantly upregulated [29]. Hvu-microRNA399 was not expressed under drought, indicating that the expression of this microRNA may be drought dependent [30].

The natural stress responsiveness of plants varies across different cultivars of the same species. Ferdous et al. used four different barley cultivars (Commander, Fleet, Hindmarsh and WI4304) in drought stress studies. The results showed differences in the natural leaf relative water content under drought conditions and differences in microRNA accumulation [31].

With increasing data confirming the importance of microRNA-mediated regulation, we decided to search for novel barley microRNAs and further characterize them.

## 2. Materials and Methods

### 2.1. Plant Material

Spring barley plants (Rolap variety) were grown in a Conviron chamber (Winnipeg, Manitoba, Canada) at 22 °C during the day and 15 °C at night with 16-h d/8-h night photoperiods under 800 µmol light. Plants were grown in autoclaved soil mixed with sand at a 7:2 ratio supplemented with medium with micro- and macronutrients and watered up to 70% SWC (Soil Water Content) for optimum growing conditions. SWC was measured as a mass of water in soil presented as percent [%], where the mass of soaked soil sample was treated as 100% and the mass of oven-dried soil was treated as 0%. SWC of 70% was maintained by weighing the pots daily.

### 2.2. Drought Stress Protocol

After plants reached the flag-leaf stage (39–41 developmental stage according to Zadoks cereal development code), the water supply was stopped to level of stress 30% of SWC as a mild drought, and 20% SWC as a severe drought. After reaching the level of 10% SWC, the plants were rewatered to 70% SWC, and the material was collected after 6 h. Plants were collected in 3 biological replicates, and samples were collected for each stress control.

### 2.3. Small RNA Libraries

Small RNA libraries were prepared for five developmental stages of barley plants (1, 2, 3, and 6 weeks and 68 d of development) and drought stress. Libraries were constructed for all developmental stages and drought experiments in three biological replicates.

Small RNA libraries were prepared as stated in [32]. RNA was enriched in small RNAs using a protocol published in [14]. In brief, RNA was extracted twice with 38% phenol solution saturated with 0.1 M sodium acetate (Roti Aqua Phenol, Roth, Karlsruhe, Germany), supplemented with 0.8 M guanidine thiocyanate, 0.4 M ammonium thiocyanate, 0.1 M sodium acetate, 5% glycerol, 0.5% sodium lauroylsarcosine, and 5 mM EDTA. To remove polysaccharides, the Ambion Plant RNA Isolation Aid (Life Technologies, Carlsbad, CA, USA) was used during phenol extraction. Next, three phenol/chloroform and two chloroform extractions were performed. RNA was precipitated in the presence of glycogen (Thermo Fisher Scientific, Waltham, MA, USA) using 1.25 vol. of ethanol and 0.5 vol. of 0.8 M sodium citrate in a 1.2 M sodium chloride solution.

RNA quality was tested using an Agilent 2100 Bioanalyzer and RNA 6000 Nano Assay (Agilent Technologies, Santa Clara, CA, USA). The RNA integrity number (RIN) value was consistently higher than 7.5. Ten micrograms of total RNA were subjected to 15% PAGE 8 M urea electrophoresis. The small RNA fraction was excised and eluted. RNA was ligated with 3′ and 5′ adapters in a two-step procedure. cDNA was synthesized using Superscript II Reverse Transcriptase (Invitrogen, Carlsbad, CA, USA) followed by PCR amplification: 98 °C for 30 s; 15 cycles of 98 °C for 10 s, 60 °C for 30 s, 72 °C for 15 s and terminated by 72 °C for 10 min. PCR products were separated electrophoretically (6% PAGE), and the cDNA representing small RNAs was excised, eluted, and quantified using a Tecan Infinite M200Pro Plate reader spectrophotometer, NanoQuant Plate (Tecan) and Quant-iT^TM^ PicoGreen^TM^ dsDNA Assay Kit (Thermo Fisher Scientific, Waltham, MA, USA).

Deep-sequencing data analysis was carried out using a previously described procedure [33].

### 2.4. Degradome Libraries

Degradome library preparation was performed according to previously published protocols [33,34]. The final libraries of 26-27-mer 5′ ends of the 5′-end phosphorylated mRNAs were sequenced by Fasteris (Plan-les-Ouates, Switzerland) on Illumina HiSeq 2500 using the Parallel Analysis of RNA ends (PARE) approach. The adapters were trimmed with the cutadapt program (minimum overlap = 19), and only reads longer than 13 nt with identified adapters were selected for subsequent analyses.

### 2.5. Isolation of Barley Flower Organs

Unfertilized spikes were harvested, and the flower organs were dissected under the highest magnification of a Leica M60 binocular (Figure 1). First, thin forceps were used to detach the lateral spikelets located on the sites of the floret. Then, the lemma and palea were carefully separated, revealing other organs: ligule, caryopsis, and stamen. Along with these organs, the rachis (on which florets are attached) and the whole inflorescence were collected. Each organ was immediately transferred to a 15-mL falcon tube submerged in liquid nitrogen. After collecting the appropriate amount of material, the tissue was transferred to Eppendorf tubes, homogenized with a handheld microhomogenizer, and RNA was isolated according to the procedure described below.

### 2.6. RNA Isolation

RNA was isolated as described previously. The concentrations and quality of all samples were measured using a Nanodrop DeNovix DS-11+ spectrophotometer and assessed on a 1.2% agarose gel. Generally, the obtained amount of RNA ranged from 30 to 40 μg per 100 mg of ground plant tissue.

### 2.7. Northern Hybridization

Two types of Northern hybridization were performed in these studies. The choice of the technique was dictated by differences in hybridization sensitivity. Our preliminary studies prompted us to perform the hybridization with TBE buffer and UV crosslinking as a quantitative method and perform a relatively more sensitive hybridization with MOPS buffer and chemical crosslinking as a qualitative method [35]. To analyze the expression of miRNAs in drought stress, total RNA was run on a 15% urea PAGE gel with 7 M urea and 10% TBE in 1× TBE buffer under 300 V. RNA was transferred to an Amersham HybondTM-N+ membrane and crosslinked with UV exposure at 1200 KJ.

For the detection of less abundant microRNAs, total RNA was run on a 15% PAA gel with 7 M urea and 20 × MOPS in 1 × MOPS buffer under 300 V. Then, RNA was transferred to an Amersham HybondTM-NX membrane and chemically crosslinked during 2 h of incubation at 55 °C with 1-methylimidazole, 1 M HCl and EDC N-(3-dimethylaminopropyl)-N’-ethylcarbodiimide hydrochloride.

Hybridization was carried out overnight at 42 °C in hybridization buffer containing 10% SDS, 1 M Na2HPO4 and 1 M NaH2PO4 with ^32^P labelled probes specific for the analyzed miRNA and for U6 snRNA as a control. After overnight incubation, the membranes were washed and exposed for two days, and screens were scanned with a FUJIFILM FLA-5100 image analyzer. The intensity of the obtained bands was calculated using Multi Gauge V2.2 software (Tokyo, Japan). All signals corresponding to microRNAs were normalized to the U6 snRNA levels.

### 2.8. TaqMan Analysis of MicroRNA Levels

Templates for all reactions were prepared according to the manufacturer’s protocol (TaqMan™ MicroRNA Reverse Transcription Kit, Applied Biosystems, cat. no. 4366596) with a use of 5× specific reverse transcription microRNA primer and 10 ng of template (TaqMan™ MicroRNA Assay cat. no. 4440886). Amplification was carried out with TaqMan™ Universal Master Mix II with UNG (Applied Biosystems, cat. no. 4440038) using the Applied Biosystems 7900HT Fast Real-Time PCR System. The results from all samples were normalized to the level of U6 snRNA.

### 2.9. RT-qPCR

Three μg of DNA-free RNA was reverse-transcribed with SuperScript III Reverse Transcriptase (Invitrogen, Carlsbad, CA, USA) and oligo(dT)_15_ (Novazym, Poland) primer. cDNA samples were diluted 4-times and 1μL was used as a template. qPCR was performed with Power SYBR^®^ Green PCR Master MIX (Applied Biosystems, Warrington, UK) on 7900HT Fast Real-Time PCR System (Applied Biosystems) in 10 μL reaction volumes in 384-well plates.

The barley ADP-ribosylation factor 1-like [GenBank: AJ508228.2] gene fragment of 61 nt was simultaneously amplified and detected as an internal reference.

### 2.10. RACE PCR

The template for 5′- and 3′-RACE experiments was prepared after combining equal amounts of RNA from 5 stages of barley development. All procedures were carried out according to the manufacturer’s protocol from the SMARTer PCR cDNA Synthesis Kit (cat. 634923). PCRs were performed with the Advantage 2 PCR Enzyme System (Clontech, Mountain View, CA, USA). Products were cloned into the pGEM T-Easy vector (Promega, Madison, WI, USA) and sequenced. The obtained sequences were aligned to the barley Barke cultivar genome in the IPK database (http://webblast.ipk-gatersleben.de/barley_ibsc/).

### 2.11. Bioinformatics and Statistical Analyses

Bioinformatical analyses included BLASTN and BLASTP comparisons against National Center for Biotechnology Information (NCBI) databases (http://ncbi.nlm.nih.gov). Secondary structures of miRNA precursors were predicted using the Folder Version 1.1 program (http://www.ncrnalab.dk/rnafolder/ with RNAfold). Alignments of all cDNAs obtained from the RACE experiments were performed using MAFFT alignment and NJ/UPGMA phylogeny software (https://mafft.cbrc.jp/alignment/server/). The sRNA sequencing data are available in the NCBI Sequence Read Archive database under accession number PRJNA526135. Bioinformatic analysis of the degradome data was performed as described in [33]. PARE RNA sequencing data are deposited in the NCBI Sequence Read Archive database under accession number PRJNA526135. The identification of the target mRNAs was performed using specific parameters: degradome score consisting of raw and normalized reads and the position in the ranked cleavage sites of a given cDNA due to cutting power and compliance score with values in the range from 0 to 18, where negative points were assigned for any sequence mismatches. The ranking of the cleavage sites was created based on cutting force and lowest compliance score.

### 2.12. Accession Numbers

All data are available in the National Center for Biotechnology Information (NCBI) Sequence Read Archive database under accession number PRJNA526135.

## 3. Results

### 3.1. Identification of Novel Barley MicroRNAs

To identify novel barley microRNAs, we prepared small RNA libraries from five stages of barley development (1-, 2-, 3-, and 6-week-old and 68-day-old plants) and performed Illumina sequencing. After the sequencing reaction, we were able to obtain more than 15 million unique, quality-filtered and adapter-trimmed reads. The BLASTn program [36] was used to align reads from our libraries to known microRNAs, and the data were further processed as described in [33]. We confirmed the presence of putative microRNAs and found that generally during the barley development, miR168-3p and miR1432-5p levels increase while the 5′U-miR156-5p level decreases (with exception for the 2-week-old barley) [32]. In our data, 21-nt long reads were the dominant fraction of the newly identified microRNA sequences (Figure 2, left panel). Presence of novel microRNAs was confirmed with Northern hybridization (Figure 2, right panel). In the case of one predicted microRNA (hvu-x13) that is expressed at very low level, it was necessary to apply a modified Northern approach—higher amount of RNA and a chemical crosslinking of RNA to the membrane [35]. Ultimately, we identified 6 novel microRNAs that we termed microRNAhvu-x9a, microRNAhvu-x9b, microRNAhvu-x11, microRNAhvu-13, microRNAhvu-x8 and microRNAhvu-x10. Analysis revealed that microRNAhvu-x9a and -x9b represent a family due to sequence similarities and have a common target gene. Moreover, microRNAhvu-x11 is a homologue of the microRNA tae-miR9662b-3p previously reported in *Triticum aestivum* [37].

### 3.2. Pri-miRNA and MIR Gene and Structure Analysis

The identified novel microRNA sequences were BLAST searched against barley cDNA sequences available at www.blast.ncbi.nlm.nih.gov/Blast.cgi, www.webblast.ipk-gatersleben.de/barley_ibsc, www.plants.ensembl.org/Hordeum_vulgare. Hairpin structures were obtained using Folder Version 1.11 BETA software, and structures with the lowest scores of Gibbs (∆G) free energy were taken under consideration (Figure 3). All identified pri-miRNAs form a hairpin structure in which microRNA and its microRNA* are located. Sequences of all microRNA* were present in the sRNA next-generation sequencing (NGS) data (Appendix A). These analyses showed that microRNAhvu-x8 and -x10 were processed from the same precursor but formed separate clusters in the sRNA NGS data (see Figure 2 and Figure 3).

To elucidate the structures of novel pri-miRNAs, we performed 5′- and 3′-rapid amplification of cDNA ends (RACE) experiments. The length of the pri-miRNA was calculated on the basis of the longest pri-miRNA 3′- and 5′-RACE products. To demonstrate that the longest pri-miRNA 5′ and 3′ ends belonged to the same precursor molecule, we carried out RT-PCR for all 6 pri-miRNAs using primers designed for the 5′ and 3′ ends of the longest pri-miRNA RACE products (Appendix A).

The length of the identified pri-miRNAs varied between 500 and 1111 nucleotides. To evaluate a given *MIR* gene structure, pri-miRNA sequences were aligned to the barley genome (https://webblast.ipk-gatersleben.de/barley_ibsc/), and the number and size of introns in the case of three *MIR* genes (*MIRHVU-X9A, MIRHVU-9B, AND MIRHVU-X13*) were identified. The novel miRNAs derived from intron-containing genes were encoded in exons. All introns within these *MIR* genes were of the U2 type. Pre-miR stem-and-loop structures were found in the second exon of *MIRHVU-X9A* and *MIRHVU-X13* genes (Figure 3).

### 3.3. Expression Pattern of Mature MicroRNAs

NGS data analysis on the microRNA level revealed that among all tested developmental stages in all cases, the highest abundance of mature microRNAs occurred on the 68th day of barley development (Figure 4 and Appendix A when spikes and flowers were formed. Therefore, with TaqMan probes, we analyzed the accumulation of novel microRNAs in seven flower organs (rachises, lateral spikelets, lodicules, caryopses, lemmas, paleas, and stamens) and in whole spikes, roots, stalks, and leaves. Each novel microRNA displayed a unique expression pattern (Figure 5 and Appendix A).

MicroRNAhvu-x11, hvu-x8, hvu-x10 and hvu-x13 were expressed at low levels throughout the plant but were highly accumulated in particular flower organs. MicroRNAhvu-x11 accumulated in the caryopses, while microRNAhvu-x10 and hvu-x8 exhibited high levels in the lodicules. MicroRNAhvu-x13 was characterized by high expression in the lodicules, rachises and roots. Another microRNA, hvu-x9b, was characterized by moderate expression throughout the plant and accumulated in the lemmas, while miRNAhvu-x9a was expressed at high levels in organs such as the leaves or spikes but was poorly expressed in flower organs (Figure 5). Taken together, these results suggest the involvement of these microRNAs in the development and function of flower organs and indicate a further direction of study.

### 3.4. Target Analysis for Novel MicroRNAs

For all novel microRNAs, target mRNAs were identified. When analyzing the degradome data and identifying target genes, we considered the presence of the mRNA cleavage site exactly between the residues paired to nucleotides 10 and 11, counting from the 5′ end of a given hybridizing microRNA sequence. For further analysis, we chose targets with cleavage sites in a proper position, with the highest number of normalized reads in the first position or with one of the first positions in the ranked cleavage sites in a given cDNA cluster. Figure 6 shows T-plots for the identification of slicing sites for a given target mRNA.

The microRNAhvu-x9a and -x9b target genes are MLOC_57686.2 = HORVU7Hr1G121200.5, characterized by the presence of NB-ARC, P-loop NTPase and winged helix-like DNA-binding domains. NB-ARC is a signaling motif shared by plant disease-resistance proteins. MicroRNAhvu-x11 targets the mitochondrial transcription terminator factor (MLOC_56533.1 = HORVU6Hr1G005650.12), which is consistent with previous findings in *Triticum aestivum* [37]. The target gene for microRNAhvu-x13 is a transcript encoding a protein belonging to the serine protease class containing the tetratricopeptide (TRP) domain (MLOC_80127.2 = HORVU6Hr1G059720.1). MicroRNAhvu-x8 targets mRNA for cleavage: MLOC_60197.2 = HORVU4Hr1G008820.1, encoding a protein with a homeobox-like and zinc finger domain and containing SWIRM domain. For microRNAhvu-x10, we identified one mRNA target, MLOC_26843.2 = HORVU0Hr1G022020.1, that is annotated as a protein containing a CS domain involved in protein-protein interactions.

### 3.5. Novel MicroRNAs and their Targets Respond to Drought

After characterizing microRNAs and their targets, we analyzed their response to various stresses. All of the identified novel microRNAs responded strongly to drought, especially to severe drought (20% soil water content (SWC). All microRNAs were downregulated in the sRNA NGS results. The results of the response of microRNA hvu-x9a, hvu-x9b and hvu-x11 to drought were confirmed with Northern hybridization, while for the other microRNAs, we either could not distinguish between microRNAs (as in the case of microRNAhvu-x10 and microRNAhvu-x8) or we were not able to detect microRNAs using the classical Northern approach (hvu-x13) (Appendix A). We then analyzed the expression of their targets by performing RT-qPCR with primers flanking cleavage sites and observed their high upregulation, especially in severe drought conditions, which suggests the involvement of microRNAs and their targets in response to this stress (Appendix A). Figure 7 summarizes the levels of microRNAs and their targets under the control and drought conditions. Taken together, these data show a clear regulation of target mRNA levels by microRNAs in response to drought.

## 4. Discussion

In this work, we report the identification of six novel barley microRNAs, their expression through barley organs and developmental stages (1, 2, 3, 6 week and 68th day) and evidence of their involvement in the response of barley to drought stress. We observed that from all, analyzed by us stages of barley development, the 68th day is the one in which novel microRNAs display highest levels of expression. This observation does not exclude the possibility that the expression level of novel barley microRNAs is higher any time between 6 weeks of barley development and 68th day. We found that microRNAhvu-x11 was previously reported as specific to *Triticum aestivum*. Here, we identified the presence of this microRNA in barley as well [37]. Moreover, microRNAhvu-x13 was also recently reported as a barley drought tolerance-associated microRNA. Additionally, bioinformatics analysis indicated several potential targets [38]. Zhou et al. showed that this microRNA was upregulated under drought conditions. However, in our experimental results, we found a different target mRNA (as mentioned in previous chapter, this is MLOC_80127.2 = HORVU6Hr1G059720.1) for this microRNA, and we observed the downregulation of this microRNA under 20% SWC (48 h) conditions. These differences may result from the fact that we tested plants subjected to drought stress after 48 h of withholding water in the case of all novel microRNAs studied, while Zhou et al. studied plants with prolonged responses to drought (5 d).

For all novel microRNAs, we have experimentally identified their target mRNAs based on degradome data analysis. The target genes for microRNAhvu-x9a and -x9b encode resistance proteins containing NB-ARC, P-loop NTPase and winged helix-like DNA-binding domains. Multiple proteins are containing the NB-ARC domain; however, they perform various functions [39]. Based on the amino acid sequence similarity of the target protein, we found that it is a homologue of the RPP8 (disease-resistance protein) protein from *Triticum aestivum* that is known to respond to pathogen infection, but thus far, its involvement in the response to drought stress has not been reported and its function regarding this condition is not known [39]. However, involvement of proteins containing NB-ARC domain and their regulatory microRNAs in drought stress have been widely explored on an example of microRNA482 from radish, microRNA815b from rice or microRNA211 from sorghum indicating important role of such proteins and microRNAs that control them and providing further direction for studies [31,40,41,42].

The target gene for microRNAhvu-x11 is a mitochondrial transcription terminator factor encoded within the nucleus and is a homologue of mTERF15 from *Triticum aestivum*. The function of this gene in barley is not known, however studies in *Arabidopsis* confirmed that mTERF15 is responsible for proper function of mitochondrial complex I because it is required for splicing of NAD2 pre-mRNA (NADH dehydrogenase 2) [43,44]. Further analysis by another group revealed that the perturbed function of mitochondrial complex I is associated with increased expression of *ATCYS12* gene and plants overexpressing *ATCYS12* display tolerance to drought stress by reducing reactive oxygen species. (ROS) accumulation [45]. Therefore, it is possible that mTERF15 (as it is upregulated in barley in drought stress) promotes the splicing of the NAD2 transcript and the accumulation of ROS as a natural response to stress treatment. Further studies are required, however, the regulation of microRNAhvu-x11 and its target *MTERF15* gene expression may be a promising approach for the regulation of plant response to drought.

The novel drought-responsive target identified for microRNA hvu-x8 encodes a protein containing the SWIRM domain and is a homologue of the SWI-SNF complex subunit SWI3B from *Aegilops tauschii*. This protein contains a helix-turn-helix motif, binds to DNA, and is predicted to mediate protein-protein interactions in the assembly of chromatin–protein complexes [46]. Increasing evidence shows that the overall stress responses and acclimation to environmental conditions in plants are at least in part attributable to changes in chromatin organization and the activity of proteins with the SWIRM domain, as reviewed by Mirouze and Paszkowski 2011 [47]. There are no reports regarding the role of SWI3B in drought stress in barley and there are no data, thus far, providing evidence that SWI3B mRNA levels are targeted by microRNAs in plants. Our results are the first evidence for the regulation of the chromatin remodeling complex component SWI3B via microRNA and its involvement in the drought-stress response in plants. However, the *Arabidopsis* swi3b mutant displays reduced sensitivity to ABA treatment and shows reduced expression of ABA-responsive genes. A strong connection between the ABA-stress response and drought response has been reported numerous times [48,49,50,51]. Nevertheless, further studies are required to learn more about the positive or negative effects of the SWI3B level on the plant response to dehydration.

Another novel target gene of microRNAhvu-x13 is a transcript encoding a protein containing a TRP domain that has no homologues in other plant species. However, proteins containing TRP motifs have been identified in all kingdoms [52]. The involvement of such proteins in the drought-stress response has been studied in Arabidopsis. AtCHIP is an E3 ligase consisting of three TRP repeats and a U-box domain. Variable temperature conditions induce the expression of AtCHIP. Mutants overexpressing AtCHIP have reduced sensitivity towards temperature fluctuations and dehydration [53]. We also observed the upregulation of TRP-encoding transcripts upon early drought. Rosado et al. showed that the TRP-containing protein TETRATRICOPEPTIDE-REPEAT THIOREDOXIN-LIKE 1 (TTL1) functions in the regulation of ABA and the dehydration signaling pathway [54]. In the ttl T-DNA mutant, the activation of the ERD1, ERD3, and DREB2A genes that are upregulated in wild-type plants upon ABA treatment was not detected, which indicates that TTL1 is a node in a dehydration-ABA signaling pathway [54]. The mechanisms of the function of TRP-containing proteins are not known in the drought-stress response. However, there have been numerous reports regarding microRNAs targeting proteins with the TRP domain. Yakovlev et al., 2017 showed that over 50 microRNAs in Norway Spruce target transcripts encoding proteins containing the TRP domain [55]. All of the above examples were based on bioinformatical predictions of targets. Nevertheless, this is the first experimental report showing post-transcriptional regulation of TRP domain-containing proteins by microRNAs.

## Figures and Tables

**Figure 1 genes-11-00488-f001:**
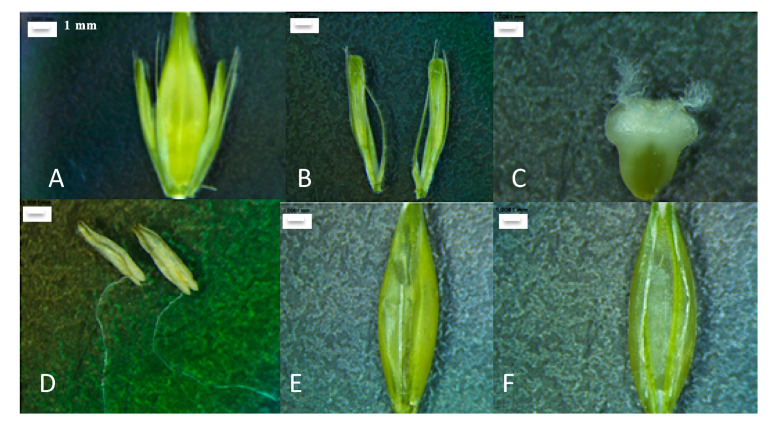
Isolated barley flower and its organs. (**A**) Barley flower. (**B**) Lateral spikelets. (**C**) Caryopsis. (**D**) Stamens. (**E**) Lemma. (**F**) Palea. Scale—1 mm.

**Figure 2 genes-11-00488-f002:**
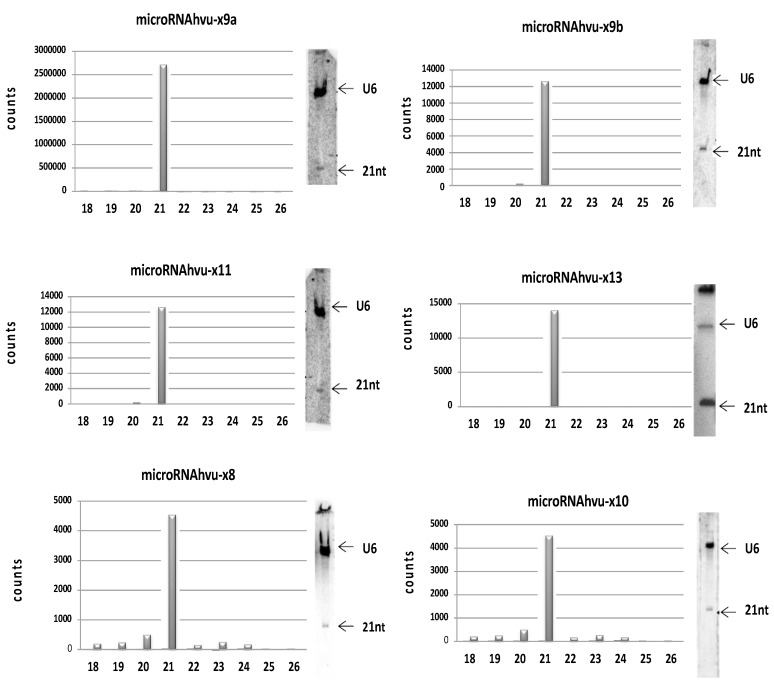
Distribution of counts for particular microRNAs in the cluster where y-axis represents normalized counts from sRNA libraries and x-axis presents length of particular small RNA in nucleotides (left panel) and validation of miRNA presence by Northern hybridization (right panel). In the case of microRNA microRNAhvu-x13 more sensitive hybridization method was used as described in Material and methods. U6 hybridization was used as loading control. Decade (Thermo Fisher Scientific) marker was used to estimate RNA fragment length. In right panels above the blots the name of each novel barley microRNA is provided. In the case of each microRNA left panel shows size distribution of small RNAs in a given cluster with the dominating cDNA fragment and mean counts from sRNA sequencing results from the 68-day-old plants.

**Figure 3 genes-11-00488-f003:**
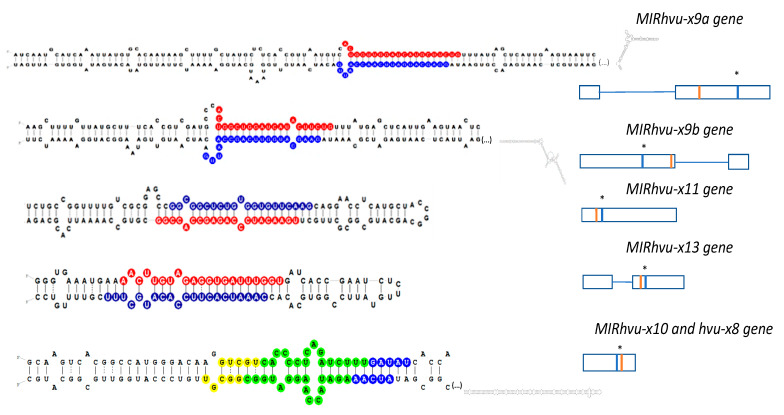
Schematic representation of identified novel miRNA precursors and their *MIR* genes. Left panel presents pre-miRNA hairpin structures of novel miRNAs. Red color indicates sequence of microRNA, the blue color indicates its corresponding microRNA*. In the case of microRNAhvu-x8 and microRNAhvu-x10 that are processed from the same precursor (microRNAhvu-x10 from 5′ arm and microRNA, hvu-x8 from 3′ arm) mature miRNAs are depicted in yellow color and miRNAs* in blue, green color indicates common nucleotides for those microRNAs and their microRNA*. Right panel shows novel *MIR* gene structures. White boxes represent exons while connecting thin lines indicate intronic sequences (1 cm corresponds to 100 bp). The red bar represents position of microRNA while blue bar represents position of microRNA*. Pre-miRNAs for miRNAhvu-x9a, -9b and miRNAhvu-x10 and -8 are very long. Dots in brackets represent sequence in the middle of hairpin structure that due to the length cannot be presented on the graph (full sequences are shown in Appendix A).

**Figure 4 genes-11-00488-f004:**
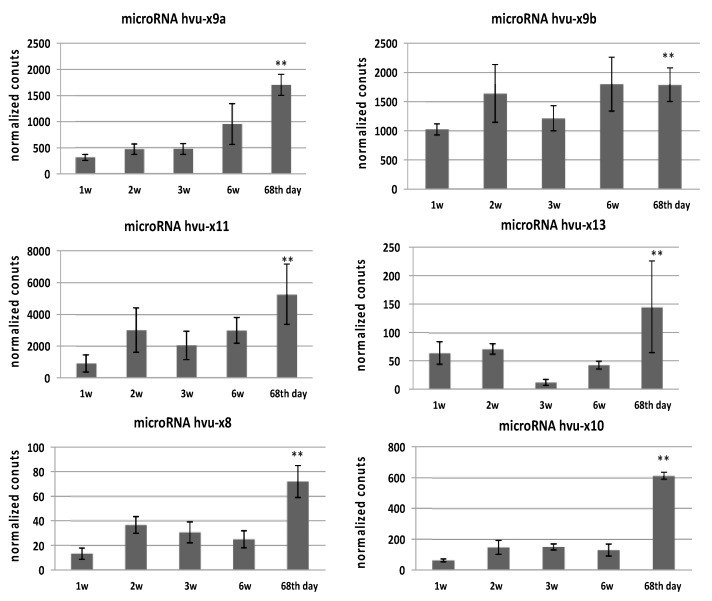
Normalized counts for novel miRNAs from sRNA libraries prepared from five barley developmental stages. Y-axis represents normalized counts from sRNA libraries and x-axis presents level of particular miRNA in barley developmental stages. Name for each novel miRNA is provided above each graph. All novel miRNAs are most abundant in the 68 d of development. Statistical significance is provided on each graph (** *p*-value ≤ 0.01).

**Figure 5 genes-11-00488-f005:**
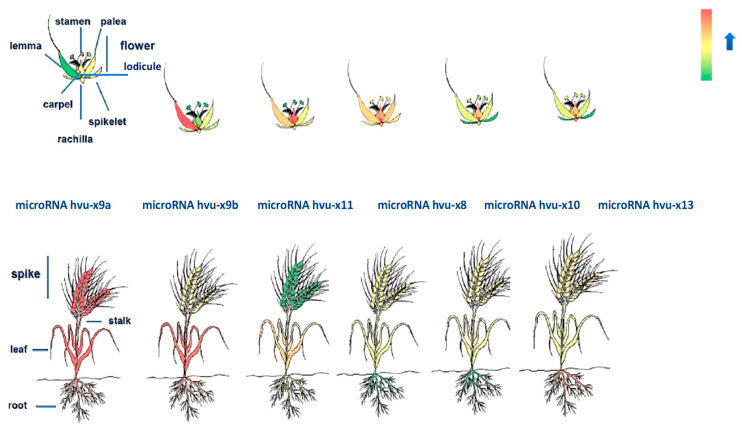
Expression pattern of novel microRNAs in barley organs. Heatmap represents expression level of novel microRNAs in particular barley organs. Red color indicates the highest expression while green represents the lowest. Top panel presents heatmaps for each novel microRNA in barley flower organs (lemma, palea, stamen, carpel, rachilla and lateral spikelets) and heatmap for microRNA hvu-x9a contains description of specific organs. Bottom panel represents expression of novel microRNAs in barley roots, leaf, stalk, and whole spike and heatmap for microRNAhvu-x9a contains description for particular organs. Above bottom panel names of each microRNA are provided.

**Figure 6 genes-11-00488-f006:**
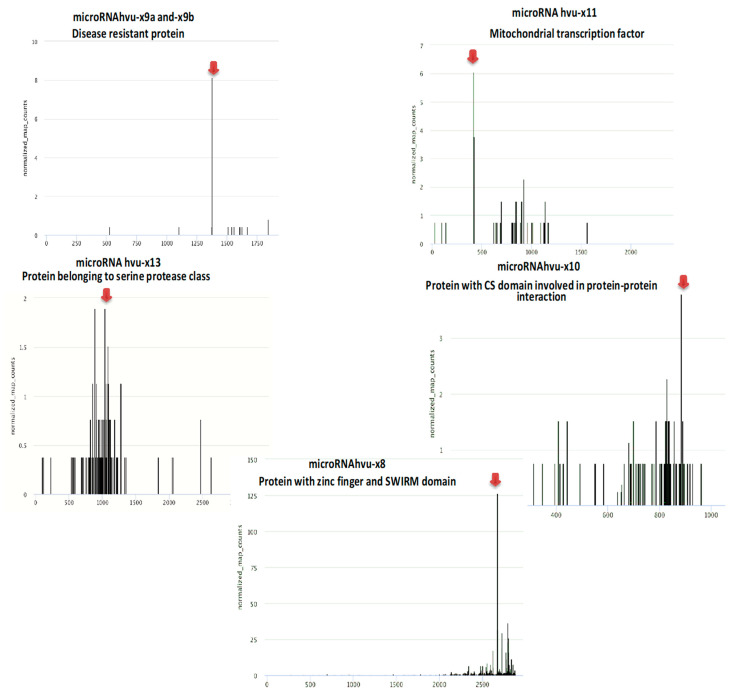
Target plots (T plots) for target mRNAs of novel barley microRNAs. The x-axis on the graphs represents the length of the targeted mRNA, y-axis shows normalized number of reads. Names of novel barley miRNAs and information about their targets are provided above the graphs. Red arrows indicate the position of the predicted cleavage site in particular mRNA.

**Figure 7 genes-11-00488-f007:**
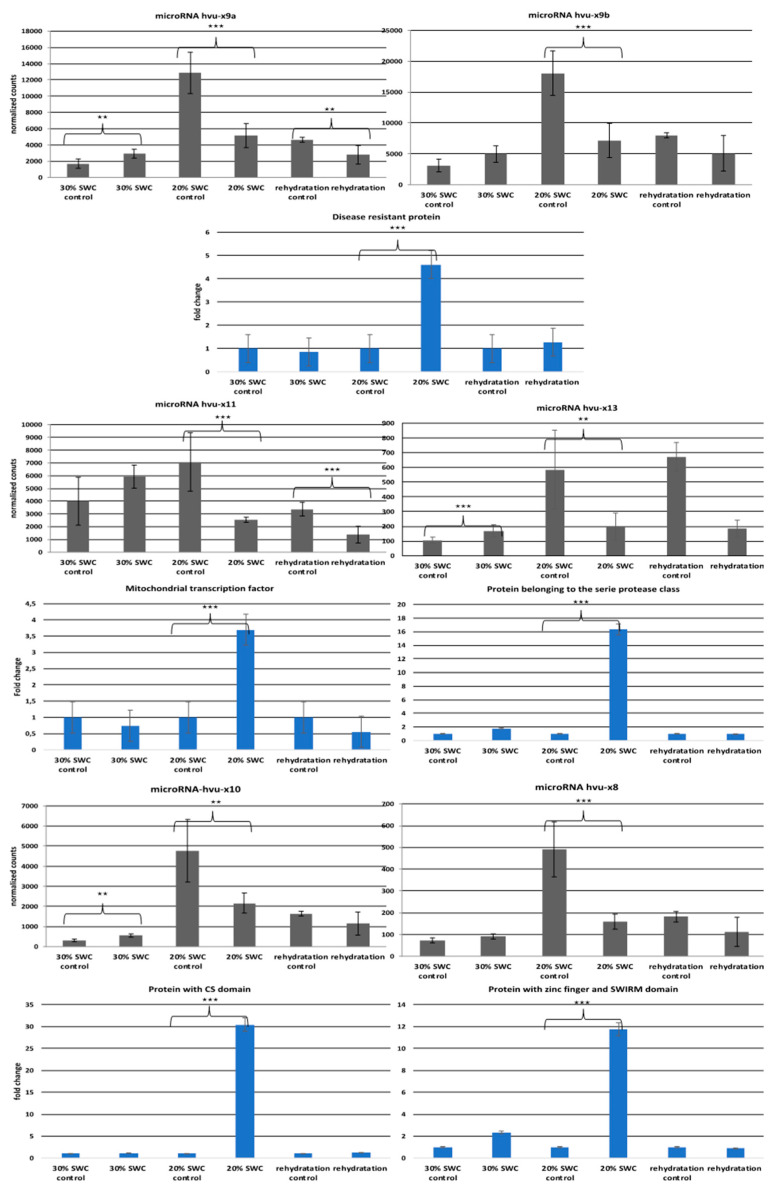
Expression analysis of novel barley microRNAs and their target genes in drought conditions (30% SWC, 20% SWC and rehydratation). Grey graphs show the level of particular microRNA based on normalized counts from sRNA libraries (OY axes points to normalized counts from sRNA libraries while OX axes point to control and different stress conditions). Names of particular microRNAs are provided above the graphs. Panels below depict blue graphs that show expression of target genes for particular microRNAs using RT-qPCR with primers flanking microRNA-recognition site (OY axes represent fold change and OX point to control and different stress conditions) ** *p*-value ≤ 0.01, *** *p*-value ≤ 0.001.

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
