# Peer review of "A Functional Network of Novel Barley MicroRNAs and Their Targets in Response to Drought"

_genes, 2020, doi:10.3390/genes11050488_

Round 1

Reviewer 1 Report

Dear Editor,

I have reviewed the article titled “A functional network of novel barley microRNAs and their targets in response to drought” by Smoczynska et al. They identified and validated six novel miRNA and found their gene targets.  The authors systematically identified and characterized new miRNAs that may have an important function in plant development and abiotic stress tolerance. The presentation and writings are of high quality and most of the claims were supported with data. However, the quality of some of the figures should be improved, and  the following concerns should be also addressed:

Introduction:

  1. The introduction requires more references.
  2. Line #29, Full name should be given for MIR before using its acronym.
  3. Line# 31, provide details about miRNA with an asterisk.
  4. Line 52-71, The authors may use abbreviation miRNA instead of microRNA for the previously characterized ones. The naming is not consistent as some places it referred to as “miRNA399” and in some places, it is mentioned as “microRNA399”. Same in Line# 82.
  5. Line#75, remove “cultivation”.
  6. Line#92-93, a similar message was given in line 74-77. Looks redundant.

Material and methods:

  1. Since this paper is highly technical, the authors should provide more details on the methods section for easy understanding. Instead of just citing a paper, the authors should include a brief description of the library preparation and small RNA enrichment. If space is a constraint, they could more the details to the supplementary information.
  2. The full name for SWC should be given in the materials and methods, when it is mentioned first time. Also, the authors measured SWC between pots for drought stress. Details should be included.
  3. Steps between lines 124-130 were not clear. If that is a PCR, how many cycles and what primers were used?
  4. In line 156, RNA isolation was not described earlier.
  5. What is PAA gel?
  6. Line 191, ipk should be IPK.
  7. Details for RT-qPCR of miRNA targets are missing. Which housekeeping gene was used and how the data was normalized?

Results:

  1. Since the authors have done small RNA sequencing, they should include a line or two about the types and levels of previously characterized miRNAs.
  2. Line#272-274: Since the authors checked the expression on only selected intervals (i.e. 1W, 2W, 3W, 6W, and 68thday), it would be difficult to say that the highest miRNA levels occurred on 68th It could have been anytime around flowering time. Thus, the statement should be modified accordingly.

Discussion:

  1. Italicize Triticum aestivum in line 365, Raphanus sativus in line 370, and Sorghum bicolor

In line 373.

  1. Discussion is lengthy and just elaborates on the predicted genes’ functions. In many cases, genes may do entirely different things than their predicted function. The authors should consider improving the discussion section.

Overall figure quality should be improved. For example, in Fig. 3., the resolution of the hairpin structures are very poor. Similarly, in fig. 6, the text is blurred and difficult to read.

Reviewer 2 Report

General comments.

The manuscript contains novel results such as

  1. The presence ofmiRNAhvu-x11 in barley
  2. The presence of a homologue of the RPP8 protein from Triticum aestivum that is known to respond to pathogen infection, but thus far, its involvement in the response to drought stress has not been reported
  3. The first report (experimental) for the post transcriptional regulation of TRP domain-containing proteins by miRNAs.
  4. Also, this manuscript firstly gives evidence for the regulation of the chromatin remodelling complex component SWI3B via miRNA and its involvement in the drought-stress response in plants.

Line 14-21. The abstract is interesting but is only around 100 words. I think the authors should add more information about the conclusions and the innovations of the manuscript, since the abstract since it usually is the most readable part of a manuscript.

Line 17. Please change analysed to analyzed

Line 22. Please add more keywords

Lines 42-44 are referred to reference 5?

Lines 60-70. It would be good if you add references about microRNA and drought stress and winter cereal e.g. wheat.

Lines 74-76. The importance of barley is that is not only that is ranking 4th in global production but also because is the main winter cereal in areas with extremely drought stress e.g. North Africa. Authors should add this information with a reference.

Line 88. Authors should change “different genotypes and cultivars of the same species.” to “across different of the same species.” because different genotyped include different cultivars.

Lines 92-93. The authors should transfer lines 77, where are referred information about barley.

Lines 92-102. Usually, at the end of the introduction is referred the purpose of the study and not the general results of the study. So authors should transfer results such as “The changes in miRNAs under drought 100 conditions were inversely correlated with the accumulation of their target mRNAs. These data depict 101 new possible metabolic pathways that shape barley plant responses to drought” to other parts of the manuscript.

Line 108. Authors should explain SWC in line 108 where is firth refered  and not in line 328.

Lines 112-113. Author write “the water supply was stopped until 30% of the SWC was reached, as a mild drought, and 20% SWC was reached, as a severe drought.”. I think that is better to write to levels of dought stress 30% SWC and 20% SWC, because 30% is writted as mild drought stress and 20% is written as severe drought stress.

Material and methods are well written.

Line 292. Authors write that “Statistical significance is provided on each graph (*pvalue≤ 0,5; **pvalue≤0,05).” The p value 0,5 is not wright.

Line 292. Please change 0,05 to 0.05.

Line 321. The number in axis X and numbers and words in axis Y are very small and not clear.

Lines 362-363. Please change the “There are multiple proteins containing” to Multiple proteins are containing
